# The Impact of Peach Rootstocks and Winter Cover Crops on Reproduction of Ring Nematode

**DOI:** 10.3390/plants13060803

**Published:** 2024-03-12

**Authors:** Sagar GC, Ivan Alarcon-Mendoza, David Harshman, Churamani Khanal

**Affiliations:** Department of Plant and Environmental Sciences, College of Agriculture, Forestry and Life Sciences, Clemson University, Clemson, SC 29634, USA

**Keywords:** cover crops, management, *Mesocriconema xenoplax*, peach, ring nematode

## Abstract

Two peach rootstocks (‘Guardian’ and ‘MP-29’) and ten winter cover crops (rye, wheat, barley, triticale, oat, Austrian winter pea, crimson clover, balansa clover, hairy vetch, and daikon radish) were evaluated in a greenhouse environment to determine their suitability to host ring nematode, *Mesocriconema xenoplax*. Each crop was inoculated with 500 ring nematodes, and the experiments were terminated 60 days after inoculation. The reproduction factor (ratio of final and initial nematode population) ranged from 0 to 13.8, indicating the crops greatly varied in their host suitability to ring nematode. ‘Guardian’ has been known to tolerate ring nematode; however, results from the current study suggest the tolerance statement is anecdotal. Another peach rootstock, ‘MP-29’, was also a good host for ring nematode, suggesting an urgency to develop ring nematode-resistant peach rootstocks. Wheat supported the least to no nematode reproduction while pea supported the greatest reproduction. The rest of the cover crops were poor to good hosts to ring nematodes. Although planting cover crops in peach orchards is not common, employing non or poor host crops can help suppress nematodes in addition to having soil health benefits. Furthermore, peach breeding programs should focus on finding and introgressing ring nematode resistance in commercial rootstocks.

## 1. Introduction

Peach [*Prunus persica* (L.) Batsch] is a high-value perennial tree fruit crop and is commercially produced in 23 US states, collectively contributing five percent towards worldwide peach production [1]. Approximately 626 thousand tons of peaches worth $651 million are produced in the US, with California, South Carolina, and Georgia being leading peach-producing states [2]. Despite peach being an economically important crop, the multimillion-dollar peach industry faces several production challenges, including nematode management. Nematodes are soil-inhabiting micro-organisms that feed on plant roots leading to poor root and plant growth, reduced ability to uptake nutrients and water, wilting and dieback, reduced quantitative and qualitative yield, and sometimes the death of the plants [3,4,5,6]. Ring nematode [*Mesocriconema xenoplax* (Raski, 1952) Loof and de Grisse 1989] is one of the major nematode species responsible for substantial damage to peach trees [3,5]. Because the damage threshold for ring nematode in peach is 20 nematodes/100 cm^3^ of soil, the presence of a relatively small nematode population in the soil can cause significant damage to peaches [7]. Both direct and indirect crop losses to nematodes are very common. Direct crop losses occur when nematodes feed on actively growing feeder roots by puncturing cells with their stylet and sucking the cellular content. It is estimated that ring nematodes can feed up to 85% of feeder roots within a year of planting [8]. The loss of feeder roots directly impacts the ability of plants to translocate nutrients and water to the upper parts. Indirect losses occur when wounds created by the nematodes provide avenue for other soil-inhabiting pathogens to enter the host. Moreover, biotic stress exerted on roots due to the nematode feeding is known to predispose trees to the bacterial canker complex known as peach tree short life (PTSL) which causes sudden limb collapse or the collapse of entire trees in the spring [8]. Despite several decades of peach breeding efforts, PTSL is still one of the leading causes of peach orchard decline and it can kill peach trees in 4 years [9], the period after which economic return of the tree is expected.

Management of ring nematode in peach is a challenge primarily due to the lack of host plant resistance in commercial peach rootstocks [10]. Although the most widely employed peach rootstock ‘Guardian’ is considered tolerant to ring nematodes, the presence of a very high population of ring nematodes in peach orchards in South Carolina and the lack of scientific studies on host status of ‘Guardian’ to ring nematode suggest the susceptibility of this rootstock to ring nematodes. Pre-plant fumigation is common in peach production; however, the efficacy of fumigants on nematodes does not last for more than a year. As the efficacy of fumigants wanes over time, the nematode population can rebound quickly to a damaging level when soil moisture and temperature become favorable for their growth and reproduction. Nevertheless, the use of fumigant chemicals is the least desirable method of nematode management due to the increasing legal restrictions on their use, negative impact on the environment and human health, and increasing cost of application [10,11,12,13]. A few non-fumigant nematicides have become available for peach production; however, their lower and short-lived effects render them less reliable than fumigants [14]. Additionally, irrigation is not a common practice in peach orchards older than three years, which makes it very difficult to apply nematicides through drip irrigation. Some biological nematicides have received label for peach, but their field performance against nematodes is not encouraging [14,15]. Although least explored, the use of cover crops can be the sustainable alternative management method for ring nematodes [11,16,17,18]. The cover crops have potential to reduce soil erosion, improve soil fertility and structure, avail feed and foliage to the livestock, and reduce the infestation of crop pests like insects, weeds, nematodes, and other plant pathogens [19,20]. Their residue can be incorporated into the soil as green manure, which can improve the production of consecutive crops by supplying nutrients along with increasing beneficial micro-organisms to the soil, enhancing crop diversity and contributing to carbon sequestration [10,11,17,18,21]. While cover cropping in peach orchards is not a common practice, finding poor to non-host crops can be of value in safeguarding peaches from ring nematodes, and promoting healthy, environmentally sound, and sustainable farming. The main objective of the current study was to evaluate the suppressive potential/host suitability of the commonly employed peach rootstocks and winter cover crops in South Carolina against ring nematode.

## 2. Results

Nematode reproduction data from the two experiments were analyzed separately because of the existence of significant experiment by treatment interactions (*p* < 0.001). The reproduction of ring nematode significantly differed among the crops as presented in Figure 1 and Figure 2. The number of nematodes per pot in the first and second experiments ranged from 0 to 960 and 240 to 10,980, respectively. Wheat did not support nematode reproduction in the first experiment, and it supported the least reproduction in the second experiment. All crops supported statistically similar nematode reproduction relative to the control ‘Guardian’ in the first experiment; however, rye, barley, triticale, oat, vetch, radish, and ‘MP-29’ supported numerically lower reproduction. Wheat, triticale, and oat in the second experiment supported significantly lower nematode reproduction relative to the control while other crops had statistically similar reproduction. Furthermore, although statistically similar, rye and barley supported numerically lower nematode reproduction while the rest of the crops supported numerically higher nematode reproduction relative to the control.

The host suitability of ring nematode significantly differed among the crops as presented in Table 1. The Rf value in the first experiment ranged from 0 to 1.9, while that in the second experiment it ranged from 0.5 to 13.8. Wheat and pea in both experiments had, respectively, the least and the greatest Rf. In the first experiment, wheat and triticale were non-hosts while rye, barley, oat, hairy vetch, and radish along with ‘Guardian’ and ‘MP-29’ were poor hosts, and pea, crimson clover, and balansa clover were good hosts to ring nematode. Similarly, in the second experiment, wheat and triticale were poor hosts, and the rest of the crops, including ‘Guardian’ and ‘MP-29’, were good hosts to ring nematode.

Plant dry biomass data from two experiments were combined for analysis because of absence of significant experiment by treatment interactions (*p* = 0.9, Table 2). The plant biomass ranged from 2.7 g to 12.8 g, with ‘MP-29’ and wheat having the least and the greatest biomasses. The biomass of ‘MP-29’ was numerically lower but statistically similar to that of the control which weighed 4.1 g. Crimson clover, balansa clover, hairy vetch, and daikon radish also had biomasses statistically similar to that of the control. Rye, wheat, barley, triticale, oat and Austrian winter pea had biomasses statistically greater than that of the control.

## 3. Discussion

Management of nematodes has become crucial as the peach industry faces new challenges that are hindering the success rate of peach orchards [22,23]. Ring nematode is of particular concern as this nematode is a major contributing factor to PTSL [5,8,24]. A very low damage threshold of ring nematode, 20 individuals/100 cm^3^ soil [7], and lack of host resistance render ring nematode management in peaches a challenge. Although the most commonly used peach rootstock ‘Guardian’ is considered resistant to ring nematodes, to our knowledge, there is a lack of scientifically established experimental data to support the resistance leaving the resistance statement merely anecdotal. A field study established by Wilkins et al. [25] in Georgia in mid 90s involving twelve peach rootstocks of which five were ‘Guardian’ selections reported ring nematode populations that were below damage threshold at planting in all rootstocks reached above damage threshold in two years of planting. Additionally, Wilkins et al. [25] monitored ring nematode for four additional years and found that the nematode population densities in all rootstocks remained above the damage threshold and were statistically similar to the commercial standard ‘Lovell’. Although not resistant to ring nematode, some of the ‘Guardian’ selections in the field evaluation by Wilkins et al. [25] had equivalent or superior horticultural qualities, making ‘Guardian’ a preferred rootstock among others. It seems like the preference of ‘Guardian’ over other cultivars for horticultural properties was passed along by the researchers and extension personnel among peach stakeholders, eventually masking the impact of ring nematode on peaches. The current study found the ring nematode population multiplying up to ten times in ‘Guardian’ within two months, suggesting the susceptibility of this rootstock to ring nematode. As crop roots in fields are exposed to ring nematodes for several months of optimal environments for nematode growth and reproduction, the nematode population density likely becomes very high during the period of active root growth. A recent field study found up to 700 ring nematodes/100 cm^3^ of soil in a peach orchard is in agreement with our statement that a very high nematode population density exists under optimal soil conditions [15]. The results from the current study as well as the previous field study [15] highlight the susceptibility of ‘Guardian’ to ring nematodes. Although it is now clear that ‘Guardian’ is susceptible to ring nematodes, the level of damage it causes on plant growth and fruit yield is unknown. Another peach rootstock, ‘MP-29’, is currently being studied as a potential replacement of ‘Guardian’ because of its better survival and vigor in the field with a history of PTSL, as reported by Reighard et al. [26]. Additionally, Beckerman et al. [27] reported ‘MP-29’ to perform better than ‘Guardian’ not only for PTSL but also for another important root disease of peach called Armillaria root rot. Shahkoomahally et al. [28] found ‘MP-29’ to induce higher soluble solids, titratable acidity, phenolic compounds, antioxidant activity, and anthocyanin content in fruits, suggesting peaches grafted on ‘MP-29’ have better fruit quality. Although ‘MP-29’ is considered to have some level of resistance against root-knot nematodes [27], studies are not available to prove its resistance against ring nematodes. Results from the current study suggest ‘MP-29’ is susceptible to ring nematodes, the susceptibility being statistically similar or greater than that for ‘Guardian’. While peach rootstocks superior to ‘Guardian’ and ‘MP-29’ in terms of horticultural performance are currently not available, peach breeding efforts must include ring nematode in their screening programs. Additional field studies are needed to determine the extent of quantitative and qualitative yield losses to ring nematodes in peaches.

The lack of host resistance and unavailability of other effective management methods have shifted the focus of some nematologists towards exploring the possibility of using cover crops as an alternative management method. The current study found wheat to be a poor to non-host, suggesting this crop can be of value in suppressing the ring nematode. While mixed results are available on the ability of wheat to suppress ring nematodes, Nyczepir and Bertrand documented wheat as being a non-host and having allelopathic effects on ring nematodes [29]. Nyczepir and Bertrand also suggested that the introduction of wheat in orchards with PTSL could be useful in reclaiming that site and replanting new peach trees when other options are not available [29]. In addition to having allelopathic effects on ring nematodes, wheat has been shown to provide other soil benefits such as an increase in soil-organic carbon, increased water retention capacity, and improved soil health through greater biological activity in the soil [30]. However, a study conducted by Nyczepir et al. [31] in the late 90s claimed wheat exhibited no suppressive effects on ring nematode population density. The same study claimed wheat of exhibiting negative effects on the growth of peach trees [31]. While the reason behind the negative growth effect of wheat on peaches is not clear, it might have resulted from a competition between wheat and peach trees for water and soil nutrients. The authors, however, stated that even though the root exudates of wheat were not attractive to ring nematodes, they did not repel the ring nematodes, implying that the use of wheat as a pre-plant ground cover management tool can be of value [31]. Further studies are needed to determine the impact of wheat on ring nematode reproduction and on peach growth.

Triticale in the current study appeared as a poor to non-host to ring nematodes, implying the employment of this crop in peach orchards can be useful for suppressing ring nematode populations. Furthermore, the suppression of ring nematode reproduction by triticale in both experiments was statistically similar to that exerted by wheat, suggesting triticale can be a good replacement candidate for wheat when the negative impact of a cover crop on peach is of concern. Nevertheless, to our knowledge, negative impacts of triticale on other crop growth have not been reported. Rather, Nyczepir et al. [29] in their early 90s study found triticale to be a poor to non-host to ring nematodes. Although further studies on the impact of triticale on ring nematodes are not available, a study conducted in mid 90s by Johnson et al. [32] found this crop to lower the population density of the southern root-knot nematode in cotton and soybean rotations.

Austrian winter pea, crimson clover, balansa clover, and hairy vetch supported greater than ten folds of nematode reproduction indicating planting these legume crops should be avoided when ring nematode is a concern. Our result was supported by the findings of Zehr et al. [33], which also stressed the legume crops as good hosts for ring nematode. Inclusion of leguminous crops provides soil benefits as they add plant available nitrogen to soils [34] thereby reducing the need of external fertilizer application. Additionally, leguminous cover crops are gaining popularity nowadays as they serve as high-quality forage in silage production [34]. Despite leguminous cover crops being superior in soil nutrient boosting and preferred silage crops, care should be taken if the crops are likely to harbor nematodes rather than suppressing them. Hairy vetch and crimson clover are also reported to be good hosts for root-knot nematodes [23] suggesting these crops would not be ideal crops in peach orchards as peaches are susceptible to both ring and root-knot nematodes.

Rye, barley, oats, and radish were found to react differently to ring nematodes in two different experiments. These crops were poor hosts in the first experiment, but they were good hosts in the second experiment. Previous studies, although only a handful are available in the literature, to determine the host status of these crops to ring nematode reported these crops to be poor or non-hosts to ring nematode. A study conducted by Nyczepir and Bertrand in the early 90s reported oat, barley, and rye to be poor to non-hosts to ring nematode [29]. A laboratory bioassay conducted by Kruger et al. in 2025 reported oat to be a poor-host to ring nematode [19]. A field study conducted by Geary et al. [34] in 2002 and 2003 to look for the potential of various Brassicaceae crops as possible alternatives to commercial chemical fumigants found radish to be very effective in suppressing ring nematodes, an indication that radish is not a good host for this nematode. Our Results from the second experiment in the current study indicating rye, barley, oats, hairy vetch, and radish to be good hosts to ring nematodes are in contrast to the findings of the previous studies, suggesting further studies are needed to determine their actual host status and to determine possible factors associated with the differential host reactions.

The reason behind differential host reactions to ring nematode by some crops (rye, barley, oats, and radish) was probably due to the existence of variable environmental conditions between two experiments. While, to our knowledge, no studies are available on the impact of the environment on the abundance, virulence and reproduction biology of ring nematode, the impact of the environment, specifically temperature, is well-documented for many plant-parasitic as well as free-living nematodes. A study published in 2023 reported that elevated soil temperatures have significant impact on the reproduction and virulence of root-knot and reniform nematodes, predicting a greater possibility of crop damage in the future era of climate change [6]. Another study found the nematode reproduction was largely a function of interactions among soil temperature, host crop and nematode species [15]. Other studies found nematode communities and their ratio were significantly changed upon minute change in the soil temperature and moisture [35,36]. We noticed the ambient temperature in the first experiment fluctuated more than in the second experiment (Figure 3). The temperature in the first experiment was high at 81 °F in the first week after inoculation, followed by a reduction in the range of 68 °F to 76 °F through the rest of the study period. The temperature in the second experiment was 68 °F on the day of inoculation, and it remained in the range of 68 °F to 73 °F until the fourth week, and 70 °F to 79 °F thereafter. The linear projection of temperatures suggested that the first experiment experienced a higher temperature in the beginning followed by a continuous decrease while the second experiment experienced the opposite phenomenon. The higher initial temperatures during the first experiment may likely have exerted stress on plants and nematodes, leading to reduced reproduction in relation to that in the second experiment.

The environment has a direct impact on soil microorganisms as well as the plants. While nematode reproduction seemed to be impacted by the ambient temperature in the current study, analysis of plant biomass suggested the lack of interactions between plant biomass and temperature, meaning it did not have a significant impact on plant biomass production. Moreover, results suggested the magnitude of the effects of temperature on nematode reproduction was greater than that on plant biomass. It is possible that the higher ambient temperature in the first experiment triggered changes in biochemical reactions in the hosts to make their roots less attractive or palatable to ring nematodes. However, understanding the mechanism behind the temperature-triggered biochemical changes in the plants is beyond the scope of the current study. Another reason behind the impact of temperature on nematode reproduction, but not on plant biomass production, could be thermo-morphogenesis. Thermo-morphogenesis is the process through which plants regulate their growth and development by regulating phytohormones in response to the fluctuations in daily temperature [37,38]. The range of temperatures during the current study period was within the optimal temperature range of the crops [39], implying that plants probably regulated their optimal growth through thermo-morphogenesis. While the current study can only speculate the thermos-morphogenesis may have existed, studies aimed at determining the impact of thermo-morphogenesis on nematode reproduction can help fill the knowledge gap and ultimately manipulate the mechanism to avoid nematode damage to crops.

While temperature was apparently the major environmental factor behind the differential reproduction of ring nematodes in two experiments, we found that relative humidity (RH) in the second experiment was 10% higher than in the first experiment (Figure 4). The linear projection of RH suggested the first experiment had an overall lower RH than the second experiment. The differences in RH may also have contributed to the differential nematode reproduction, although studies on the impact of RH on nematode reproduction are currently lacking. It would be reasonable to mention that RH has a direct impact on plant growth, which may indirectly impact nematode reproduction. However, further studies are needed to elucidate the impact of RH on nematode reproduction, survival, and virulence.

Cover crop planting is not a common practice in peach orchards. While some orchards are employing summer cover crops as ground cover, the use of winter cover crops is very rare. The employment of non or poor host cover crops can help avoid PTSL as host plant resistance to ring nematode is not available in commercial peach rootstocks and the most widely used rootstock ‘Guardian’ as well as the other available rootstock ‘MP-29’ are susceptible to ring nematodes. Moreover, the planting of winter cover crops not only helps suppress nematodes during the early reproductive phase of the peach tree, but it also helps maintain soil health by conserving soil moisture and nutrients, adding soil organic matter, and promoting beneficial microbiome growth [10,11,17,18,19,20,21]. Furthermore, the use of cover crops can lead to sustainable agriculture. Peach growers should be made aware of the cover crops that suppress a wide range of nematodes in addition to adding significant organic matter to the soil.

## 4. Materials and Methods

### 4.1. Preparation of Nematode Inoculum

Nematode inoculum was collected from a peach orchard previously known to be infested with *Mesocriconema xenoplax* at Musser Fruit Research Farm of Clemson University. Soils were collected using a 2.5 cm diameter and 30 cm long foot-powered conical soil sampler from weed-free zones under the tree canopy at a depth of 15–20 cm. Ring nematodes were extracted from the soil using the centrifugal-flotation technique [40].

### 4.2. Establishment of Experiments

Two temporally spaced greenhouse experiments were established in six-inch top-diameter plastic pots to evaluate two peach rootstocks, ‘Guardian’ and ‘MP-29’, and ten commercially available cover crops against ring nematode. The cover crops employed in this study represent the commonly planted winter cover crops in South Carolina, and include rye (*Secale cereale* L., cv. Wrens Abruzzi), wheat (*Triticum aestivum* L., cv. Saluda), barley (*Hordeum vulgare*), triticale (×*Triticosecale* Wittmack)), oat (*Avena sativa* L., cv. Coker 227), Austrian winter pea (*Pisum sativus* subsp. *arvense*), crimson clover (*Trifolium incarnatum* L.), balansa clover (*T. michelianum* Savi.), hairy vetch (*Vicia villosa* Roth), and daikon radish (*Raphanus sativus* cv. Longi-pinnatus). The cover crop seeds were purchased from Adams-Briscoe Seed Company (Jackson, GA, USA). Two-month-old ‘Guardian’ seedlings and ‘MP-29’ cuttings were obtained from the Clemson University Musser Fruit Research Farm. ‘Guardian’ was used as a control. Each pot received 1.5 kg of sandy loam soil steam-sterilized for four cycles at 123 °C for 45 min. Following two days of pot filling, each pot was planted with ten seeds of a cover crop or a two-week-old peach seedling. The experiment was established as a randomized block design with five replications. Soils were infested after two weeks of plantation/seeding by pipetting an aqueous suspension containing 500 ring nematodes into three depressions arranged into a triangular pattern, 0.5 cm diam. × 5 cm deep [41]. Cover crops were thinned to 5 plants per pot one day before inoculation. Standard fertilization, watering, and insect management practices were conducted. Each experiment was terminated two months after inoculation. Soil from each pot was placed in a walk-in cooler at 4 °C and processed within 72 h of collection. The nematodes were extracted from a subsample of 100 cm^3^ of soils using the centrifugal flotation technique [40]. The nematodes extracted were enumerated within 24 h of extraction using a stereoscopic microscope (Martin Microscope Company, Easley, SC, USA) at 40× magnification. Plant materials were dried at 55 °C for two weeks and biomass was recorded. Temperature and relative humidity data during the study periods were also recorded.

### 4.3. Reproduction Factor and Host Status

The ratio of the final nematode population (Pf) and initial inoculum level (Pi) was used for the calculation of the reproduction factor (Rf) [42]. Each crop was designated as a non-host (Rf ≤ 0.1), poor host (0.1 < Rf < 1), and good host (Rf ≥ 1) based on its Rf value [43].

### 4.4. Data Analysis

Data were subject to one-way analysis of variance using R-stat version 4.2.2 [44]. Data from each experiment were analyzed separately when significant experiment-by-treatment interactions were present. Before analysis, data were checked for normality and any non-normal data were transformed following the R package bestNormalize function. Nematode reproduction data in the first experiment were normally distributed, but not in the second experiment. Therefore, the nematode reproduction data in the second experiment were transformed using the square root function, and Rf data in the same experiment were transformed using the arc sine function. Plant biomass data in both experiments were normally distributed. Tukey’s HSD test (*p* ≤ 0.05) was used for *post hoc* mean comparisons. Treatment means presented in tables represent untransformed values.

## Figures and Tables

**Figure 1 plants-13-00803-f001:**
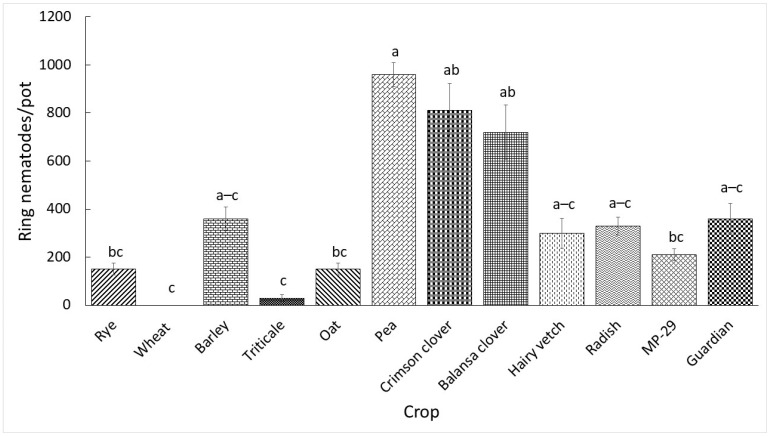
Ring nematodes per pot 60 days after inoculation with 500 ring nematodes in the first greenhouse experiment. ‘Guardian’ was used as control. Data are means of five replications. Means followed by a common letter across bars are not significantly different according to Tukey’s HSD test (*p* ≤ 0.05).

**Figure 2 plants-13-00803-f002:**
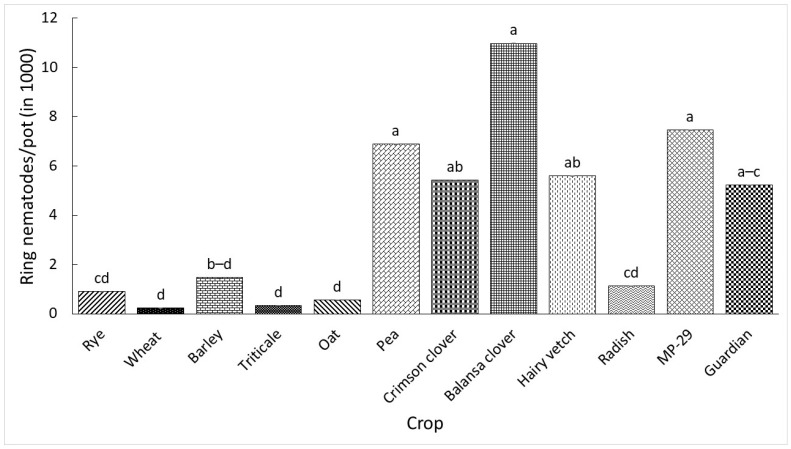
Ring nematodes per pot 60 days after inoculation with 500 ring nematodes in the second greenhouse experiment. ‘Guardian’ was used as control. Data were means of five replications. Means followed by a common letter across bars are not significantly different according to Tukey’s HSD test (*p* ≤ 0.05).

**Figure 3 plants-13-00803-f003:**
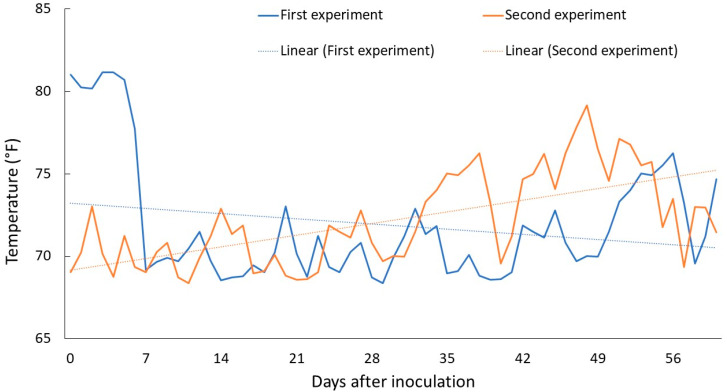
Temperature (°F) during the first and second greenhouse study period.

**Figure 4 plants-13-00803-f004:**
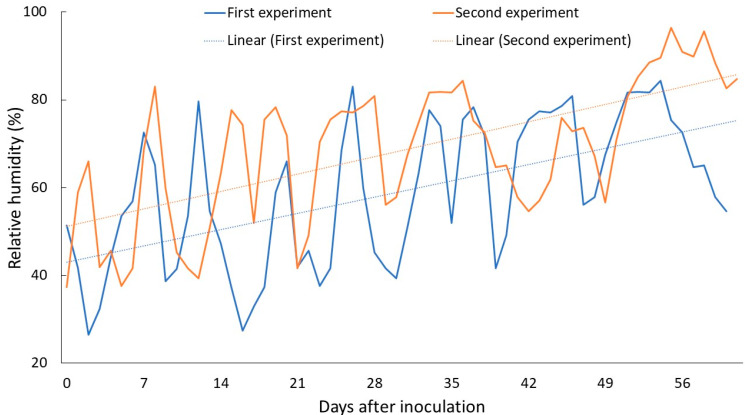
Relative humidity (RH%) during the first and second greenhouse study period.

**Table 1 plants-13-00803-t001:** Reproduction factor (Rf, ratio of final nematode population and initial inoculum level) of ring nematode on two peach rootstocks and ten cover crops and their host status at two months post-inoculation in a greenhouse environment.

Crop	First Experiment	Host Status	Second Experiment	Host Status
Rye	0.3 ± 0.05 bc	PH	1.8 ± 0.53 cd	GH
Wheat	0.0 ± 0.00 c	NH	0.5 ± 0.52 d	PH
Barley	0.7 ± 0.10 a–c	PH	2.9 ± 0.65 b–d	GH
Triticale	0.1 ± 0.03 c	PH	0.7 ± 0.52 d	PH
Oats	0.3 ± 0.05 bc	PH	1.1 ± 0.69 d	GH
Austrian winter pea	1.9 ± 0.10 a	GH	13.8 ± 1.54 a	GH
Crimson clover	1.6 ± 0.23 ab	GH	10.9 ± 0.84 ab	GH
Balansa clover	1.4 ± 0.23 ab	GH	22 ± 1.70 a	GH
Hairy vetch	0.6 ± 0.13 a–c	PH	11.2 ± 0.97 ab	GH
Daikon radish	0.7 ± 0.08 a–c	PH	2.3 ± 0.62 b–d	GH
‘MP-29’	0.4 ± 0.05 bc	PH	14.9 ± 0.73 a	GH
‘Guardian’	0.7 ± 0.13 a–c	PH	10.4 ± 1.96 a–c	GH
*p*-value	<0.0001	-	<0.0001	-

‘Guardian’ was used as control. Plants were inoculated with 500 ring nematodes. Data are means of five replications. Means followed by a common letter within a column are not significantly different according to Tukey’s HSD test (*p* ≤ 0.05). Values presented as mean ± standard error. Host status was designated based on mean Rf values. NH = non-host, PH = poor host, GH = good host.

**Table 2 plants-13-00803-t002:** Plant dry biomass at two months post inoculation with 500 ring nematodes in a greenhouse environment.

Crop	Dry wt. (g)
Rye	11.8 ± 1.57 ab
Wheat	12.8 ± 1.81 a
Barley	10.1 ± 1.2 a–c
Triticale	12.1 ± 0.85 a
Oats	7.6 ± 0.37 b–d
Austrian winter pea	10.8 ± 0.79 ab
Crimson clover	4.9 ± 0.47 d–f
Balansa clover	3.3 ± 0.38 ef
Hairy vetch	6.0 ± 0.74 c–e
Daikon radish	5.4 ± 0.42 d–f
‘MP-29’	2.7 ± 0.16 f
‘Guardian’	4.1 ± 0.56 ef
*p*-value	<0.0001

‘Guardian’ was used as control. Data were combined over two experiments as means of ten replications. Values are presented as mean ± standard error. Means followed by a common letter within a column are not significantly different according to Tukey’s HSD test (*p* ≤ 0.05). Plant materials were dried at 50 °C for two weeks.

## Data Availability

The datasets of the current study will be available from the corresponding author on reasonable request.

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
