# Peer review of "The Impact of Peach Rootstocks and Winter Cover Crops on Reproduction of Ring Nematode"

_plants, 2024, doi:10.3390/plants13060803_

Round 1
Reviewer 1 Report
Comments and Suggestions for Authors
Authors present interesting results on important nematode pest of peach trees. Presented results are inconsistent to some extent as different data were obtained in experiment which was repeated twice, however authors are discussing this honestly in relevant part of the manuscript.
Some minor remarks:
l. 335 was the 500-specimen final inoculum per pot?
Dry mass of experimental plants was apparently established however you do not mention this in Material and methods; please add paragraph on that containing temperature and time of drying.
If the Guardian rootstock is regarded as control you should designate this variant eg as Guardian (control) in figures and place results od this variant on the left or right side of the graph.
I would remove the last sentence in the paragraph of discussion dealing with triticale, it is mainly fodder crop so information presented in this place is surplus.
Reviewer 2 Report
Comments and Suggestions for Authors
Dear Authors,
The authors undertook a very interesting research topic in the context of searching for impact of peach rootstocks and winter cover crops on reproduction of ring nematodeVery interesting research was showed. Unfortunately, the authors did not avoid a few factual errors, but this does not diminish their value in the preparation of the manuscript.
Overall, combining the first results and then discussion made it very hard to follow the results and to further understand the functional relevance of these results. I recommend disentangling the three sections: methodology, result and discussion.
Lack of conclusions.
There is a problem with reading the article and looking for tables or charts. A bit of chaos in this manuscript.
